# Pain and Function in the Runner a Ten (din) uous Link

**DOI:** 10.3390/medicina56010021

**Published:** 2020-01-07

**Authors:** Peter Francis, Isobel Thornley, Ashley Jones, Mark I. Johnson

**Affiliations:** 1Department of Science and Health, Institute of Technology Carlow, Carlow, Ireland; 2Musculoskeletal Health Research Group, Leeds Beckett University, Leeds LS13HE, UK; i.thornley@leedsbeckett.ac.uk (I.T.); Ashley.D.Jones@leedsbeckett.ac.uk (A.J.); m.johnson@leedsbeckett.ac.uk (M.I.J.); 3Centre for Pain for Research, Leeds Beckett University, Leeds LS13HE, UK

**Keywords:** chronic pain, pain management, Achilles, tendon, running

## Abstract

A male runner (30 years old; 10-km time: 33 min, 46 s) had been running with suspected insertional Achilles tendinopathy (AT) for ~2 years when the pain reached a threshold that prevented running. Diagnostic ultrasound (US), prior to a high-volume stripping injection, confirmed right-sided medial insertional AT. The athlete failed to respond to injection therapy and ceased running for a period of 5 weeks. At the beginning of this period, the runner completed the Victoria institute of sports assessment–Achilles questionnaire (VISA-A), the foot and ankle disability index (FADI), and FADI sport prior to undergoing an assessment of bi-lateral gastrocnemius medialis (GM) muscle architecture (muscle thickness (MT) and pennation angle (PA); US), muscle contractile properties (maximal muscle displacement (Dm) and contraction time (Tc); Tensiomyography (TMG)) and calf endurance (40 raises/min). VISA-A and FADI scores were 59%/100% and 102/136 respectively. Compared to the left leg, the right GM had a lower MT (1.60 cm vs. 1.74 cm), a similar PA (22.0° vs. 21.0°), a lower Dm (1.2 mm vs. 2.0 mm) and Tc (16.5 ms vs. 17.7 ms). Calf endurance was higher in the right leg compared to the left (48 vs. 43 raises). The athlete began a metronome-guided (15 BPM), 12-week progressive eccentric training protocol using a weighted vest (1.5 kg increments per week), while receiving six sessions of shockwave therapy concurrently (within 5 weeks). On returning to running, the athlete kept daily pain (Numeric Rating Scale; NRS) and running scores (miles*rate of perceived exertion (RPE)). Foot and ankle function improved according to scores recorded on the VISA-A (59% vs. 97%) and FADI (102 vs. 127/136). Improvements in MT (1.60 cm vs. 1.76 cm) and PA (22.0° vs. 24.8°) were recorded via US. Improvements in Dm (1.15 mm vs. 1.69 mm) and Tc (16.5 ms vs. 15.4 ms) were recorded via TMG. Calf endurance was lower in both legs and the asymmetry between legs remained (L: 31, R: 34). Pain intensity (mean weekly NRS scores) decreased between week 1 and week 12 (6.6 vs. 2.9), while running scores increased (20 vs. 38) during the same period. The program was maintained up to week 16 at which point mean weekly NRS was 2.2 and running score was 47.

## 1. Introduction

Achilles tendinopathy (AT) is the 2nd most common running injury [1,2]. It is primarily located in the mid-portion of the tendon or at the insertion with the calcaneus (insertional tendinopathy) [3]. The stages of tendinopathy have been characterized as reactive (a non-inflammatory cellular response to acute loading that results in tendon thickening), dysrepair (failed healing that demonstrates greater matrix breakdown and the separation of collagen) and degenerative (increased cell death, matrix breakdown, neuronal and capillary in growth) [4]. Pain can be present or absent at any stage of the tendon pathology continuum. Furthermore, tendons can have discrete regions at different stages of the pathology-continuum at the same time [4,5]. Pathological findings as a result of imaging the tendon do not always correlate with patient symptoms [6]. 

The majority of running injuries are characterized by repetitive overuse and are usually to tissues ill-equipped for load absorption (i.e., non-contractile) [1]. The insidious onset of pain at a threshold (e.g., <5/10 on a Numeric Rating Scale; NRS) [7] below that experienced during traumatic insults often allows the runner to continue to run. In other words, there is a dissociation between pain and function in chronic injuries [8]. There are several pharmacological (e.g., NSAID’s, injection therapy) and non-pharmacological (e.g., heavy isometric loading, shock-wave therapy, cross-training) aids that produce short-term analgesic effects in tendinopathy [9,10,11,12,13]. However, chronic pain can reach a threshold whereby it is more akin to that experienced in traumatic pain (>5/10 on an NRS). This can force the athlete to stop running and seek longer-term resolutions.

The strongest evidence for the long-term resolution for Achilles tendon pain comes from exercise protocols [14]. Variations of the Alfredsson protocol (straight and bent knee eccentric exercise on the edge of a step) [15,16] appear to be most effective in patients with mid-portion tendinopathy. Excessive dorsi-flexion on the edge of a step is thought to increase the compressive load at the tendon insertion and therefore, straight leg exercises on a flat surface appear to be more effective for those with insertional tendinopathy [17]. These protocols have been further enhanced by a greater understanding of the other neuromuscular deficits that occur as a result of Achilles tendon injury. For example, externally paced eccentric exercise using a metronome standardizes the time the muscle is under tension and enhances neural plasticity [18].

The reasons pain and function respond positively to this form of training are multi-factorial [19]. To the best of the authors’ knowledge, there are no scientific reports which capture self-reported function (Victoria institute of sports assessment–Achilles questionnaire; VISA-A, the foot and ankle disability index FADI, and FADI Sport) [20,21,22], muscle architecture (Ultrasonography; US), muscle contractile properties (Tensiomyography; TMG), muscle endurance (Calf Endurance Test) and pain (Numeric Rating Scale; NRS) pre and post conservative Achilles tendon management.

## 2. Case Report

### 2.1. Case History

The patient provided written informed consent for the publication of this case report. A male runner (30 years, 10-km time: 33 min 46 s) had been running with suspected insertional AT for 110 weeks (~2 years) having had the symptoms for the previous 5 years. The average miles per week for the 2-year period was 30. The average miles per week during phases of race preparation was ~40 and the maximum miles per week was 50. During this period, pain (~≤5/10; NRS) and or stiffness was present every morning during the first steps on walking. Pain usually subsided as running progressed and returned after periods of prolonged rest. The runner used a 4-day-per-week running schedule and a 3-day-per-week cross-training schedule to manage running loads applied to the tendon and to lower the risk of other injury. Occasionally, pain increased (≥5/10; NRS) and was relieved using heavy double-legged isometric exercise (~85 kg) and reducing running from four days to three. No structured Achilles tendon rehabilitation was undertaken other than calf raise exercises on days the runner attended the gym (maximum 2*per week). The above summarises the training and pain status of the runner for much of the 2 years prior to cessation. This period included personal bests over 5 km (16:10), 10 km (33:46) and half-marathon (78:00).

Nine weeks before cessation (week 101/110), the runner began to increase the volume and sometimes intensity of one weekly run in an attempt to train for a marathon. This elevated daily pain to 7/10 and to a point where conservative self-management (as above) was no longer effective. The runner attended a sports medicine physician for an ultrasound guided high-volume stripping injection as prescribed in Humphreys et al. [10]. The diagnosis of right-sided medial insertional AT was made by a qualified medical doctor using ultrasonography. The doctor had ~20-years’ experience in the diagnosis of musculoskeletal conditions, diagnostic ultrasound and steroid injection therapy. The doctor was a member of the British Association of Sports and Exercise Medicine. Diagnosis was obtained by palpating the painful site on the medial calcaneus and using ultrasonography to image the same area (Figure 1). Over the course of 1 week (no running), the runner failed to respond to injection therapy and ceased running.

### 2.2. Assessment

The runner reported to the laboratory (Figure 2) at Leeds Beckett University and completed the Victoria institute of sports assessment-Achilles questionnaire (Victoria institute of sports assessment-Achilles questionnaire; VISA-A), the foot and ankle disability index (FADI and FADI Sport), prior to undergoing an assessment of bi-lateral gastrocnemius medialis (GM) muscle architecture (muscle thickness (MT) and pennation angle (PA); US), muscle contractile properties (maximal muscle displacement (Dm) and contraction time (Tc); Tensiomyography (TMG)) and calf endurance (40 raises/min).

VISA-A and FADI scores were 59%/100% and 102/136, respectively. Compared to the left leg, the right GM had a lower MT (1.60 cm vs. 1.74 cm), a similar PA (22.0° vs. 21.0°), a lower Dm (1.2 mm vs. 2.0 mm) and Tc (16.5 ms vs. 17.7 ms). Calf endurance was higher in the injured leg (right) compared to the left (48 vs. 43 raises).

### 2.3. Intervention

The athlete ceased running for 5 weeks and began a metronome guided (15-BPM), 12-week progressive eccentric training protocol (3 sets, 12 reps, 2*day) on a flat surface, using a weighted-vest (1.5 kg increments per week). The exercise was performed by bilaterally plantar flexing both legs and lowering unilaterally on the affected leg to a count of 4 s (15 BPM). The athlete also received six sessions of shockwave therapy (2000 shocks per session at a pressure and frequency of 1.8–2.2 bar and 10 Hz, respectively) concurrently within the first 5 weeks. The athlete recorded daily pain scores (NRS) [23] throughout the intervention (Figure 3). On return to running (after 5 weeks), the athlete also recorded a score composed of miles * rate of perceived exertion (RPE) [24]. Running miles* rate of perceived exertion (RPE) was recorded in order to provide a crude estimate of running load. Running began with a 2.5-mile jog, barefoot and on a hard grass surface (British Summer Time). This was progressed (week 6–13) until the athlete could run 8 miles, three times per week at an RPE of 11 (light). The laboratory measures were re-assessed at the 12-week point. The athlete continued to perform the eccentric training protocol and record pain and running load data up to week 16 (Figure 3). Between week 13 and 16, the three 8-mile runs were replaced with an 8-mile run over an undulating surface (shod; RPE: 13), a track session that began with 4 × 400 m and progressed to 16 × 400 m (shod; RPE 15) and a long run that progressed to 12 miles (barefoot and occasionally shod; RPE 11).

### 2.4. Outcomes

The athlete had a compliance of 80.4% (135/168 possible sessions) to the 12-week eccentric training protocol. At the 12-week point, foot and ankle function improved according to scores recorded on the VISA-A (59% vs. 97%) and FADI (102 vs. 127/136). Figure 4 displays pre and post left–right differences in gastrocnemius muscle architecture and contractile properties. 

In the injured leg, improvements in MT (1.60 cm vs. 1.76 cm) and PA (22.0° vs. 24.8°) were recorded via US. Improvements in Dm (1.15 mm vs. 1.69 mm) and Tc (16.5 ms vs. 15.4 ms) were recorded via TMG. Calf endurance was lower in both legs and the asymmetry between legs remained (L: 31, R: 34). Pain intensity (mean weekly NRS scores) decreased between week 1 and week 12 (6.6 vs. 2.9), while running scores increased (20 vs. 38) during the same period. The program was maintained up to week 16 at which point mean weekly NRS was 2.2 and running score was 47 (Figure 5).

## 3. Discussion

The runner in this case report ran for ~2 years with persistent pain and achieved several personal bests. This demonstrates, in this athlete at least, that pain can be managed in a way that allows a high level of athletic function. This underscores the title of this paper which suggests that pain and function are not as closely linked in chronic pain as they may be in pain resulting from traumatic insult. In fact, baseline results of the calf endurance test appear to demonstrate a compensatory mechanism whereby calf endurance in the injured leg was higher than that of the non-injured [25]. Calf endurance, and to a lesser extent PA, may have helped to compensate for the asymmetry in MT and Dm reported. 

This athlete did not respond to a high-volume stripping injection, which may suggest that this type of intervention is not as effective in insertional tendinopathy as it appeared to be for mid-portion tendinopathy [10]. It may also reflect intra-individual responses to injection therapy. Overall, pain and function appeared to respond positively as a result of the combined shockwave therapy and eccentric loading protocol administered over the 16-week period. It is difficult to know from a case report whether the effects demonstrated were due to one or both interventions. It should be noted that pain did not begin to lower on a consistent basis until week 7, and all shockwave therapy had been administered by week 5. These findings, combined with the rise in function and fall in pain from ~week 10, suggest that the eccentric loading protocol was the dominant factor in recovery. This suggestion is underpinned by the substantial improvements in neuromuscular function reported.

In the injured leg (right), there were improvements in gastrocnemius medialis MT (~10%) and PA (~12.7%), as measured by US, which appeared to be reflected by the relative improvement in Dm (~47%) and, to a lesser extent, Tc (~6.7%). These findings were expected, as they mirror improvements in muscle size and function in response to resistance exercise [25]. Strength increases prior to changes in muscle size and is usually proportionally greater, i.e., modest gains in muscle size are usually associated with 3–4 times the relative strength gains [26,27]. In the early stages of resistance training, this is suggested to be as a result of an increase in the neural drive to muscle [28]. Häkkinen and colleagues reported the largest motor units to be recruited within the first 7 weeks of resistance training [29]. The force output of a muscle increases in response to the number of sarcomeres in parallel. An increase in PA facilitates more sarcomeres in parallel, i.e., an increase in physiological cross-sectional area (CSA) without a change in anatomical CSA [30]. It is likely that the increase in PA and the overall increase in contractile mass (MT) contributed to the increase in Dm reported in this athlete. However, Dm is not a measure of force but of stiffness and therefore, it is likely to be heavily influenced by the tendon aponeurosis complex and intramuscular connective tissue. These are the structures which transfer muscle force to the bone. Increases in musculotendinous stiffness have previously been reported in response to resistance training [31,32] and likely contribute to changes reported in Dm. These changes in connective tissue stiffness are thought to be primarily driven by an increase in type I collagen synthesis [33]. 

This interpretation of an increase in Dm representing an increase in stiffness is somewhat at odds with researchers in the field who suggest a lower Dm to be associated with a greater muscle stiffness [34]. These interpretations are based primarily on cross-sectional studies comparing power and endurance athletes [35] and founded on what remains a limited understanding of the physiology that underpins measures obtained from Tensiomyography. In our laboratory, we are still optimizing the standard operating procedures [36,37] in order to obtain stable measurements prior to a more in-depth interpretation of what the data might indicate physiologically. Our experience from this case report and on-going investigations in our laboratory suggest that differences in Dm are dependent on the context in which they are reported. We suggest that in the case of cross-sectional comparisons, as mentioned above, lower Dm is indeed indicative of higher muscle stiffness. However, in the case of training or detraining we suggest that the increase or decrease in contractile mass and PA can lead to an increase in or decrease in Dm. We advise caution in the interpretation of these findings and others, as interpretation of the data obtained from Tensiomyography remains in its infancy. 

The modest reduction in Tc might be expected from an intervention that did not challenge the speed of muscle contraction. The improvements reported are likely due to an increase in the number of sarcomeres in series [38] and the increase in PA, which would allow sarcomeres to operate closer to their optimum length [25]. Calf endurance appeared to decline during the 12-week period, which we suggest is due to the initial 5 weeks without running and the gradual buildup of running up to week 12. It is likely that the removal of a stimulus that had been there for ~2 years led to this reduction. The asymmetry between legs remained, suggesting that some of the adaptations which occurred in the painful leg over time remained. 

There are several limitations we would like to acknowledge related to the interpretation of this case report. Firstly, although tendon structural abnormalities are present on ultrasound, this cannot be inferred to represent pathology [39]. In fact, the link between findings on imaging and pain in runners is weak for several musculoskeletal conditions [40,41,42]. That being said, the presence of loading specific pain and pain on palpation suggest an accurate diagnosis was received. Secondly, although we suggest the eccentric exercise protocol was the primary driver behind improvements demonstrated in this case, the influence of the injection and the shockwave therapy cannot be ruled out. Recent evidence suggests that high-volume image-guided injections, as used in this case, demonstrate improvements in pain and function with or without steroid in mid-portion AT [43]. This did not appear to be the case in this report. Although pain did not decrease on a consistent basis until week 7, a significant dose of shockwave therapy was administered during the first 5 weeks. Furthermore, there are promising results in terms of medium and long-term follow-up in mid-portion AT following shockwave therapy [44]. There are several mechanisms by which shockwave is speculated to have an effect on pain and tissue healing. The short-term effects on pain are thought to occur due to hyper stimulation analgesia [45]. In the medium term, it has been suggested that a sharp rise in substance P release and a subsequent long decrease may lead to longer lasting relief from pain [46]. Tendon repair is speculated to be promoted post shockwave therapy via an increase in tenocyte proliferation and an induction of growth factors [47]. Finally, tendinopathy is a condition associated with chronic pain. Disability levels associated with chronic musculoskeletal pain are more closely associated with cognitive and behavioral aspects of pain than sensory and biomedical ones. Positive outcomes are associated with changes in psychological distress, fear avoidance beliefs, self-efficacy in controlling pain and coping strategies [48]. Therefore, in this case, the break from running, and the knowledge of a combined shockwave therapy and exercise intervention coupled with an expected timeline of improvement may have had just as big an impact as the intervention itself.

## 4. Conclusions

This athlete ran at a high level for 2 years with persistent Achilles pain. Even after an evidence-based rehabilitation programme and a gradual return to running, this athlete was never entirely pain-free. The findings in this runner appear to underscore the need for greater awareness among clinicians about the dissociation between pain and function in Achilles tendinopathy. Pain education and limiting fear avoidance may be as important as physical rehabilitation. The neuromuscular adaptations in this study were substantial and the physiology which underpins them is multifactorial. To what extent these neuromuscular adaptations contribute to the reduction in pain and increase in running specific function is unknown and it will perhaps always be an almost impossible research question to answer. However, it is likely that the combination of neuromuscular adaptations and structural adaptations within the tendon itself help to better balance stress on biological tissue preventing overload of a specific portion of tendon. 

## Figures and Tables

**Figure 1 medicina-56-00021-f001:**
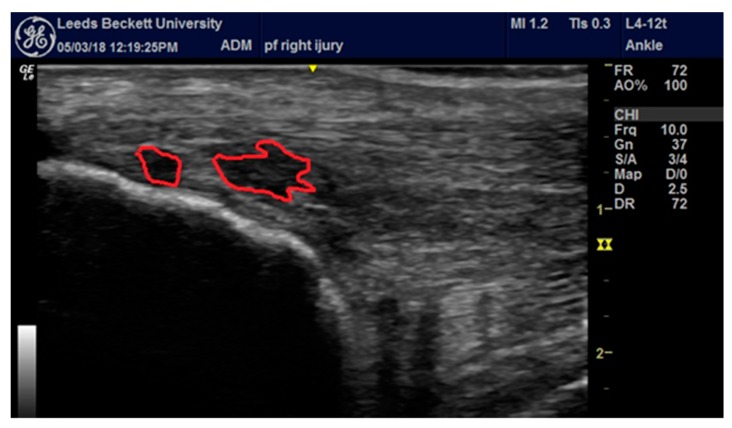
Identification of right-sided medial insertional tendinopathy.

**Figure 2 medicina-56-00021-f002:**
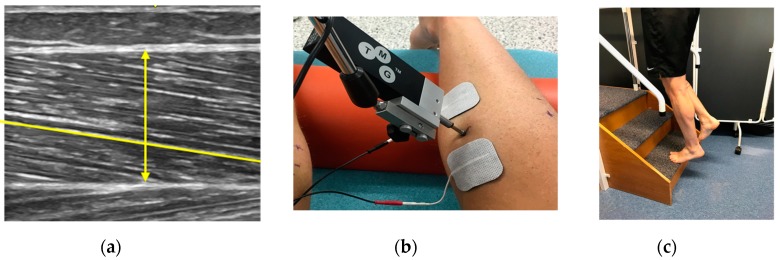
The use of ultrasonography (US), tensiomyography (TMG) and the calf endurance test in the assessment of (**a**) muscle architecture, (**b**) contractile properties, and (**c**) endurance.

**Figure 3 medicina-56-00021-f003:**
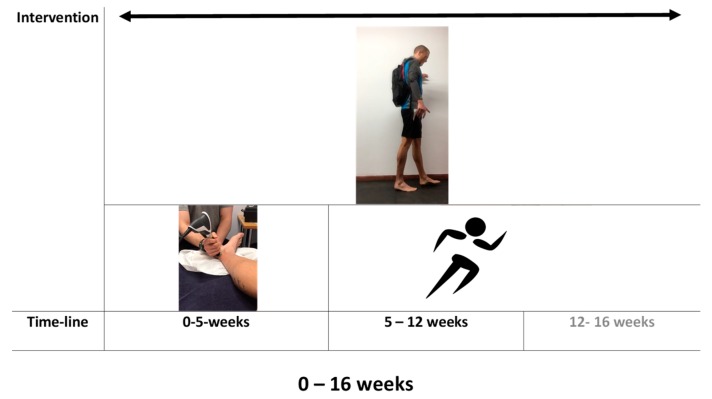
The intervention for the management of medial insertional Achilles tendinopathy.

**Figure 4 medicina-56-00021-f004:**
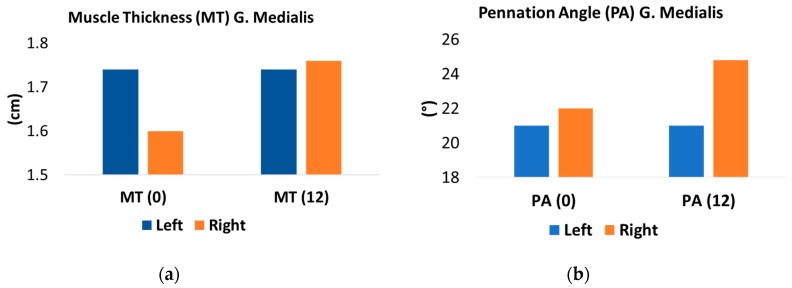
Changes in gastrocnemius muscle architecture ((**a**) muscle thickness and (**b**) pennation angle) and contractile properties ((**c**) muscle displacement and (**d**) contraction time) pre and post intervention.

**Figure 5 medicina-56-00021-f005:**
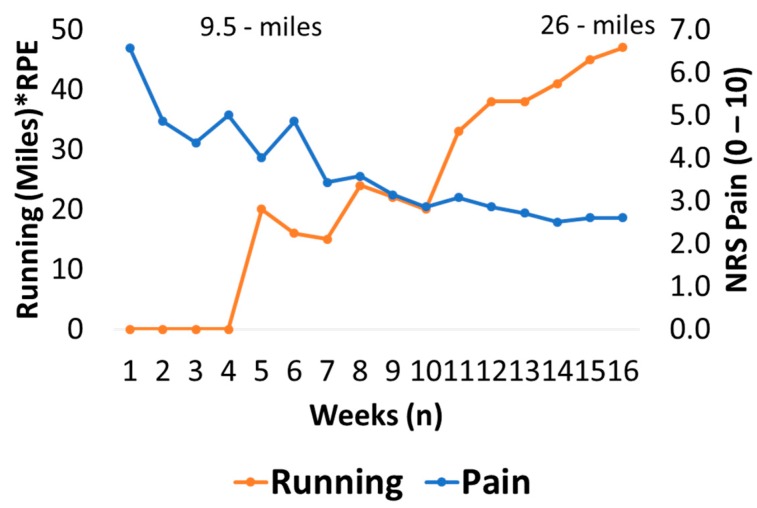
Weekly pain (numeric rating scale; NRS) and function (miles*rate of perceived exertion; RPE) response to intervention. Weekly mileage totals began with 9.5 on week 5 and ended with 26 on week 16.

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
