# Peer review of "Pain and Function in the Runner a Ten (din) uous Link"

_medicina, 2020, doi:10.3390/medicina56010021_

Round 1

Reviewer 1 Report

Thanks for your effort, the manuscript has been revised and improved, a good report on an interesting topic

Reviewer 2 Report

Thank you for the opportunity to re-evaluate the manuscript, in relation to the proposed changes, all of them have been considered and complement the excellent work previously presented.

This manuscript is a resubmission of an earlier submission. The following is a list of the peer review reports and author responses from that submission.

Round 1

Reviewer 1 Report

First of all I would like to thank the authors for the report of this interesting clinical case. 

This study is a single case report of a patient diagnosed with Achilles Tendinopathy. The manuscript is well presented and documented based on current literature and available scientific knowledge about the treatment of this type of condition. Although many quality clinical trials have already been reported with the precedents used, the contribution of muscle structure measures is innovative in this clinical case, and provides an interesting perspective for the rehabilitation of these patients.
However, there are some aspects that should be clarified by the authors in order to consider the manuscript for publication.

1) How was the diagnosis of Achilles tendinopathy made? should be exposed in the presentation of the case.

2) The ultrasound image provided as proof of the presence of enthesopathy should be discussed based on findings that ultrasound imaging tests may not be relevant to the clinical diagnosis or outcome of tendinopathies (PMID: 26390270).

3) In the 12-week treatment period there are 5 weeks in which the patient ceases training, in that period he received shockwave treatment sessions. It is necessary to better explain the reason for this therapeutic choice as the first line of treatment, as it is not considered in the introduction as one of the usual procedures for these processes. And to discuss further based on the effects of the technique reported in the literature (PMID: 23728271) on the possible action of this technique on the reported variables of muscular structural change.

4) From week 10 to 16 the intensity of the pain is maintained in mild-moderate (3 points in NRS), it should be discussed if it is considered that other variables related to the perpetuation of pain, may be the cause for which the patient maintains these levels despite increased tolerance to the load, for example some of a psychosocial nature (PMID: 26407586).

Thank you for the opportunity to review this magnificent work.

Reviewer 2 Report

Thank you for your contribution, the case report is well written and results and interventions are properly described.

The aim of the paper is not completely clear whether it is more focused on the efficacy of a self-administered training protocol or on the pathophysiology of tendinopathy.

It is difficult to address the improvement seen in the athlete to the exercise program or to sohockwave treatment (or, more likely,to both interventions). The discussion is focused on changes in parameters mesuared with TMG and those data,as the authors stated, are difficult to interpret.

I didn't find any comment on pharmacological treatment eventually taken by the patient (NSAIDs?) during the intervention period.
